# Acetaldehyde breath test as a cancer risk marker in patients with esophageal and hypopharyngeal squamous cell carcinoma

Fumisato Sasaki[1]*, Shuji Kanmura[1], Kohei Oda[1], Hidehito Maeda[2], Masayuki Kabayama[1], Hiromichi Iwaya[1], Yuga Komaki[1], Shiho Arima[1], Shiroh Tanoue[1], Shinichi Hashimoto[1], Hiroshi Fujita[2], Akio Ido[1]

1 Department of Human and Environmental Sciences, Digestive and Lifestyle Diseases, Kagoshima University Graduate School of Medical and Dental Sciences, Kagoshima, Japan, 2 Department of Gastroenterology and Hepatology, Izumi General Medical Center, Izumi city, Kagoshima, Japan

* bungohs@m2.kufm.kagoshima-u.ac.jp

## Abstract

Patients with inactive acetaldehyde dehydrogenase 2 (ALDH2) are at high risk for esophageal squamous cell carcinoma (ESCC) and hypopharyngeal squamous cell carcinoma (HPSCC). The acetaldehyde breath test (ABT) may demonstrate ALDH2 gene polymorphisms. We evaluated the usefulness of the ABT in patients with ESCC and HPSCC. The squamous cell carcinoma (SCC) group consisted of 100 patients who were treated with endoscopic submucosal dissection (ESD) for ESCC or HPSCC, and the control group (HC) consisted of 275 healthy subjects. The SCC group comprised the "single subgroup" ($n = 63$), in which a single lesion was initially treated with ESD, and the "multiple subgroup" ($n = 31$), in which multiple lesions were initially treated with ESD. First, we compared the groups' risk factors for carcinogenesis and measured the acetaldehyde-to-ethanol (A/E) ratio. Then we tested the groups' differences in the abovementioned carcinogenic risk factors. We found that the proportion of individuals in the SCC group with inactive ALDH2 (A/E ratio $\geq$ 23.3) was significantly higher than that in the HC group ($p = 0.035$), as was the A/E ratio ($p < 0.001$). Also, the proportion of individuals with inactive ALDH2 in the multiple subgroup was significantly higher than that in single subgroup ($p = 0.015$), as was the A/E ratio ($p = 0.008$). In conclusion, ABT may be a potential screening tool for detecting people at risk of ESCC and HPSCC. In addition, it could be a useful tool in detecting patients at risk of multiple or double carcinomas among patients with ESCC and HPSCC.

**Trial registration:** Trial Registration number: UMIN000040615 [https://rctportal.niph.go.jp/en/detail?trial_id=UMIN000040615], Data of Registration: 01 46 June 2020, retrospectively registered.

## Introduction

Hypopharyngeal squamous cell carcinoma (HPSCC) and esophageal squamous cell carcinoma (ESCC) are two of the deadliest cancers worldwide [1–3]. HPSCC and ESCC are often

**Data Availability Statement:** All relevant data are within the paper and its Supporting information files.

**Funding:** The authors received no specific funding for this work.

**Competing interests:** The authors have declared that no competing interests exist.

**Abbreviations:** A/E ratio, the ratio of breath acetaldehyde level to breath ethanol level; ABT, acetaldehyde breath test; ALDH2, acetaldehyde dehydrogenase 2; ESCC, esophageal squamous cell carcinoma; ESD, endoscopic submucosal dissection; HPSCC, hypopharyngeal squamous cell carcinoma; HRA score, health risk appraisal score; LVL, Lugol-voiding lesion.

advanced when detected, the prognosis is relatively poor. [4,5]. Acetaldehyde is the first metabolite of ethanol and a definite carcinogen for organs affected by the abovementioned cancers [2,3]. Field cancerization is a biological process in which large areas of cells at a tissue surface or within an organ are affected by one or more carcinogenic alterations. The results of one study indicated that field cancerization might be due to the accumulation of acetaldehyde rather than alcohol itself [6].

Acetaldehyde is metabolized primarily by acetaldehyde dehydrogenase 2 (ALDH2). People with inactive heterozygous ALDH2 who drink alcohol are at high risk for HPSCC and ESCC [7–9]. ALDH2 activity is related to two alleles: ALDH2*1 (active ALDH2) and ALDH2*2 (inactive ALDH2). ALDH2 genotypes are classified as follows: ALDH2*1/*1 (homozygous active ALDH2); ALDH2*1/*2 (heterozygous inactive [<20% activity] ALDH2; and ALDH2*2/*2 (homozygous inactive [0% activity] ALDH2) [10–13]. Carriers of the ALDH2*2 allele (ALDH2*1/*2 and ALDH2*2/*2) account for 40%–50% of East Asian populations [14–16]. Many epidemiological studies have revealed that ALDH2*1/*2 individuals who drink large amounts of alcohol are at high risk for HPSCC and ESCC [17–21]. According to a previous report, people with ALDH2*2 variants have higher risks of head and neck and esophageal cancers, because the ALDH2 activity in their tissues is much lower compared to that in the gastrointestinal tissues of healthy people [22].

Aoyama et al. [23] recently reported the development of a new breath study, the acetaldehyde breath test (ABT), which can measure very low levels of acetaldehyde and alcohol quantitatively after ingestion of a very small amount of alcohol (100 mL of 0.5% ethanol) and can accurately and rapidly identify ALDH2*2 allele carriers. According to Aoyama et al., the ratio of breath acetaldehyde level to breath ethanol level (A/E ratio) could identify ALDH2*2 allele carriers very accurately; the A/E ratio in carriers of the ALDH2*2 allele was significantly higher than that in participants who did not have the ALDH2*2. In addition, the accuracy, sensitivity, and specificity of the A/E ratio for identifying carriers of the ALDH2*2 allele were 96.4%, 100%, and 92.5%, respectively [23].

However, there is no report on the clinical significance of ABT for patients with ESCC or HPSCC. In recent years, as a result of advances in endoscopic technology, many cases of superficial ESCC and superficial HPSCC have been discovered and treated with methods of endoscopic resection, such as endoscopic submucosal dissection (ESD) [24,25]. These cases may reflect the genesis of full-blown ESCC or HPSCC. In this study, we analyzed the clinical significance and usefulness of ABT as a disease marker for superficial ESCC and superficial HPSCC in patients treated with ESD.

## Materials and methods

### Patients and methods

Two groups of subjects were compared: patients with squamous cell carcinoma (SCC group) and healthy controls (HC group). The patients in the SCC group had undergone ESD treatment for superficial ESCC or superficial HPSCC, or both, at the Kagoshima University Hospital, Kagoshima, Japan, between December 2016 and August 2018. The criteria for inclusion in the SCC group were (1) pathological diagnosis of superficial ESCC or superficial HPSCC in the resected specimens, (2) the patient's written informed consent, and (3) the patient's ability to provide breath samples. Exclusion criteria were (1) history of surgical resection of any part of the upper gastrointestinal tract, (2) history of alcohol allergy, and (3) being younger than 20 years of age.

The patients included in the HC group were examined in a medical checkup that included esophagogastroduodenoscopy and ABT at Izumi General Medical Center, Kagoshima, Japan.

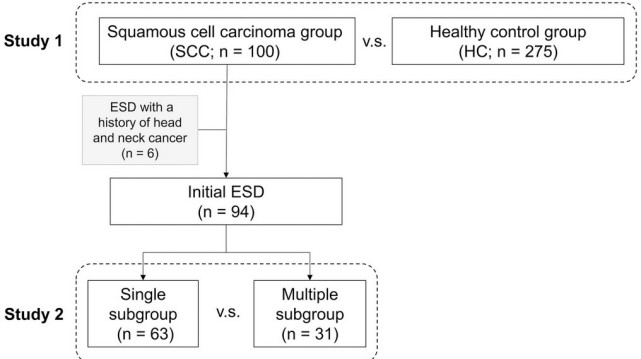

**Fig 1. Outline of this study.** The flowchart of enrollment for analysis. The "single" subgroup comprised patients in whom a single lesion was initially treated with endoscopic submucosal dissection (ESD). Meanwhile, the "multiple" subgroup comprised patients in whom multiple lesions were initially treated with ESD.

The criteria of inclusion in the HC group were (1) age of 20 years or older, (2) the patient's written informed consent, and (3) the patient's ability to provide breath samples. Exclusion criteria were (1) history of surgical resection of any part of the upper gastrointestinal tract, (2) history of alcohol allergy, (3) age below 20 years, (4) detection of cancer of the upper gastrointestinal tract by esophagogastroduodenoscopy, and (5) current pregnancy.

A total of 100 participants were included in the SCC group and 275 in the HC group (Fig 1). Table 1 lists the characteristics of the participants in the SCC and HC groups. Of the patients in the SCC group, 64 had superficial ESCC, 14 had superficial HPSCC, and 22 had both conditions. Six patients were eventually excluded because of history of head and neck or esophageal cancer; 3 had undergone oral resection of pharyngeal cancer, 1 had undergone radiotherapy for pharyngeal cancer, 1 had undergone radiotherapy for vocal cord cancer, and 1 had undergone chemoradiotherapy for tongue cancer. The remaining 94 patients were subdivided into the "single" subgroup, which comprised 63 patients in whom a single lesion was initially treated with ESD, and the "multiple" subgroup, which comprised 31 patients in whom multiple lesions were initially treated with ESD.

History regarding alcohol consumption and smoking was carefully documented in health risk appraisal (HRA) score [26], and a table of daily alcohol consumption was compiled for each age (S1 File). The HRA score is useful for the risk assessment of esophageal cancer in the Japanese population. It is calculated from the amount and frequency of alcohol consumption, smoking, flushing, and consumption of fruits and vegetables. The higher the score, the greater is the risk [26]. In addition, all patients were surveyed in detail about alcohol flushing and frequency of toothbrushing. Alcohol flushing, in which the face turns red after drinking a small amount of alcohol, has been reported to be associated with esophageal cancer [26], and the frequency of toothbrushing was recently reported to be correlated with risk for head and neck cancer [27]. We scored frequency of toothbrushing as follows: 0 points for brushing almost every morning, noon, and evening; 1 point for brushing often; and 2 points for rarely or never brushing.

Squamous dysplasia, a preneoplastic lesion, is identified easily on Lugol chromoendoscopy as a Lugol-voiding lesion (LVL) [28]. For all patients in the SCC group, the presence of LVLs on Lugol chromoendoscopy of esophagus was documented in accordance with the report of Katada et al. as grades A (no LVLs), B (1–9 LVLs), and C (≥10 LVLs; S2 File) [28].

**Table 1. Characteristics of the patients with SCC and healthy controls.**

| Characteristics | SCC (*n* = 100) | HC (*n* = 275) | *p* |
|---|---|---|---|
| Sex (male/female) | 95/5 | 167/108 | <0.001 |
| Age (years) | | | |
| <40 | 0% | 21.1% | |
| 40–49 | 1.0% | 32.4% | |
| 50–59 | 18.0% | 29.5% | |
| 60–69 | 43.0% | 13.1% | |
| 70–79 | 32.0% | 3.6% | |
| ≥80 | 6.0% | 0.4% | <0.001 |
| Mean ± SD | 66.4 ± 8.5 | 49.1 ± 10.9 | <0.001 |
| Daily alcohol consumption (g) | | | |
| <25 | 12.0% | 76.7% | |
| ≥25 | 88.0% | 23.3% | <0.001 |
| Mean ± SD | 53.6 ± 29.4 | 15.1 ± 18.9 | <0.001 |
| Smoking (yes/no) | 58/42 | 147/128 | 0.454 |
| Alcohol flushing | | | |
| Current flushing | 28.0% | 26.5% | |
| Former flushing | 24.0% | 14.2% | |
| Never flushing | 48.0% | 59.3% | 0.052 |
| Alcohol flush reaction (yes/no) | 52/48 | 112/163 | 0.053 |
| HRA score | 8.17 ± 2.46 | 3.41 ± 2.87 | <0.001 |
| Toothbrushing score | | | |
| 0–2 | 60.9% | 79.6% | |
| 3–4 | 32.6% | 20.4% | |
| 5–6 | 6.5% | 0.0% | <0.001 |
| Mean ± SD | 1.90 ± 1.35 | 1.32 ± 1.35 | <0.001 |
| Type of lesion (number of patients) | | | |
| ESCC | 64 | - | |
| HPSCC | 14 | - | |
| Overlap | 22 | - | |

ESCC, esophageal squamous cell carcinoma; HC, healthy control; HRA, health risk appraisal; HPSCC, hypopharyngeal squamous cell carcinoma; SCC, squamous cell carcinoma; SD, standard deviation.

## Acetaldehyde breath test

Participants were asked to drink 100 mL of 0.5% ethanol in one draught after at least 12 h of fasting and abstinence. The 0.5% ethanol was made with vodka, which contains little acetaldehyde [29]. Breath samples were collected with dedicated gas bags immediately before and 1 min after participants drank the alcohol. We used the dedicated gas bags to collect the end-tidal gas [23]. The gas bags were made of vinyl alcohol polymer and uniquely shaped to remove the gas derived from the physiological dead space. Approximately 100 mL of end-tidal gas can be collected with one breath into the bag. In this study, the breath was collected into these bags at standard temperature with air conditioning.

## Measurement of breath acetaldehyde and ethanol levels

Breath acetaldehyde and ethanol levels were measured by highly sensitive gas chromatography [30] performed using the AERoChrome (Nissha FIS, Inc., Osaka, Japan), which can measure

acetaldehyde and ethanol content. This device can also calculate the A/E ratio for each individual 1 min after ethanol ingestion. With the cutoff value of the A/E ratio set to 23.3, a previous study revealed that the AERoChrome was able to determine the presence of ALDH2*1 (active ALDH2) and ALDH2*2 (inactive ALDH2) with an accuracy of 96.4% [23].

## Statistical analyses

The absolute and relative frequencies for qualitative variables were calculated for the SCC and HC groups. Statistical analysis was conducted using IBM SPSS Statistics 26 software (IBM, Armonk, NY, USA) and Prism (version 6 or later; GraphPad Software, Inc., La Jolla, CA, USA). The continuous variables are expressed as means and ranges. The Mann–Whitney *U* test was employed to compare continuous data. Pearson's chi-squared test or Fisher's exact test was employed to analyze categorical data to compare proportions. A *p* value of less than 0.05 was considered statistically significant. Logistic regression analysis for univariate and multivariate analyses was used to examine the predictors of the development of SCC and multiple lesions in the esophagus and the hypopharynx. All relevant data are within the manuscript and its S3 File.

## Ethical approval

This study was approved by the institutional review board of the Kagoshima University Hospital Clinical Research Ethics Committee [Research No. 28–137] and Izumi General Medical Center. Written informed consent was obtained from all participants. The procedures followed were in accordance with the guidelines for the care and use of laboratory animals of their institution or national animal welfare committee, or with the World Medical Association's Declaration of Helsinki (1964, and its later amendments).

# Results

## SCC vs HC groups

We compared the SCC and HC groups with regard to characteristics and risk factors for carcinogenesis, such as alcohol consumption, smoking, measured A/E ratio, and proportion of subjects with inactive ALDH2 (A/E ratio $\geq$ 23.3). Patients in the SCC group were significantly older and had a higher proportion of men than did the HC group. In addition, HRA scores of the SCC group were higher than those of the HC group. There was no significant difference with regard to smoking and alcohol flushing. In the SCC group, the frequency of toothbrushing was lower (Table 1). Univariate analysis also revealed that the proportion of individuals with inactive ALDH2 (A/E ratio $\geq$ 23.3) was significantly higher in the SCC group (43%) than in the HC group (31.3%; $p$ = 0.035). In addition, the A/E ratio in the SCC group (24.0 ± 16.5) was significantly higher than that in the HC group (18.8 ± 16.2; $p$ < 0.001; Fig 2). The A/E ratio was extracted as a factor contributing to carcinogenesis in multivariate analysis, together with age, alcohol consumption, and HRA score (Table 2).

## Single versus multiple

We compared differences in the abovementioned characteristics and carcinogenic risk factors between the single and multiple subgroups. There were no significant differences with regard to gender, age, alcohol consumption, smoking, alcohol flushing, HRA score, or toothbrushing frequency. The proportion of patients in the multiple subgroup with grade C LVLs was significantly higher than that in the single subgroup. There was no difference between the two groups with regard to other organ cancers (Table 3). The proportion of individuals with inactive

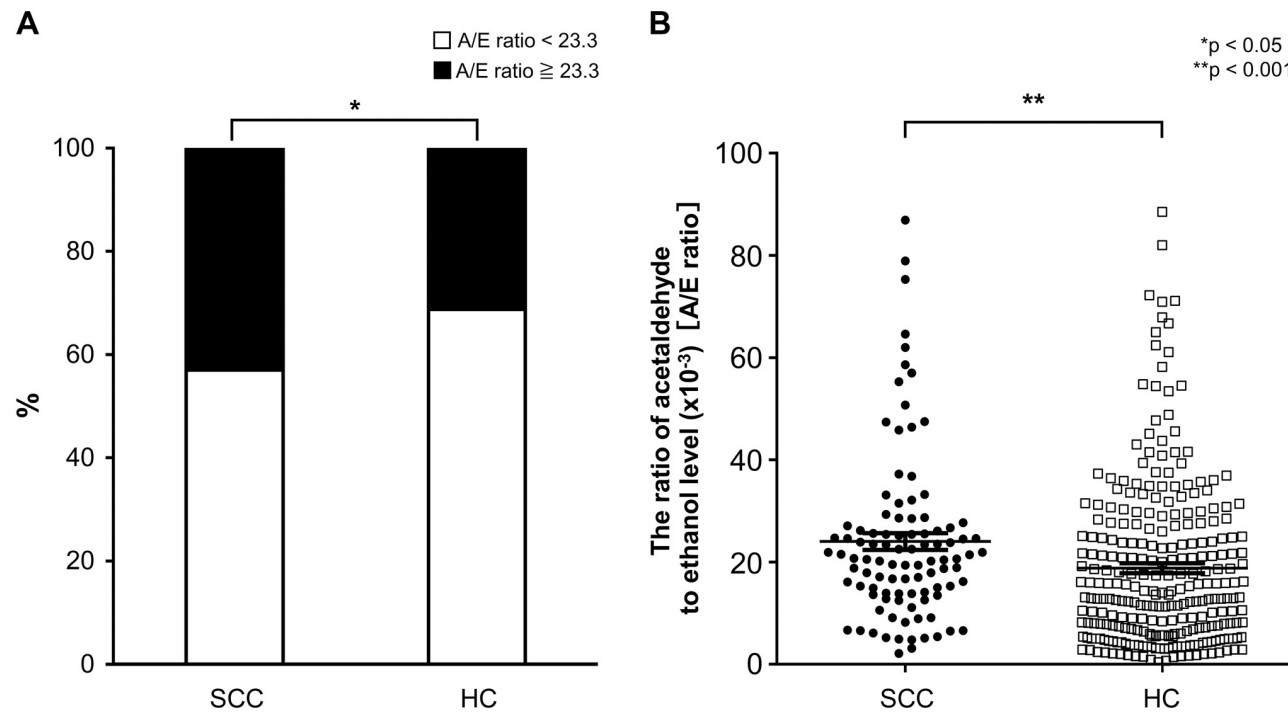

**Fig 2. Comparison of the A/E ratios between SCC and HC.** (a) The proportion of individuals with inactive acetaldehyde dehydrogenase 2 (ALDH2; acetaldehyde -to-ethanol [A/E] ratio ≥ 23.3) was significantly higher in the group of patients with squamous cell cancer (SCC; 43.0%) than among the healthy controls (HC; 31.3%; $p = 0.035$, chi-squared test). (b) Plot of the A/E ratios of each individual 1 min after alcohol ingestion. The A/E ratio in the SCC group (24.0 ± 16.5) was significantly higher than that in the HC group (18.8 ± 16.2; $p < 0.001$, Mann–Whitney $U$ test).

ALDH2 (A/E ratio ≥ 23.3) was significantly higher in the multiple subgroup (61.3%) than in the single subgroup (34.9%; $p = 0.015$). In addition, the A/E ratio in the multiple subgroup (31.3 ± 20.0) was significantly higher than that in single subgroup as well (21.2 ± 13.9; $p = 0.008$; Fig 3). In multivariate analysis, the A/E ratio and LVL grade were extracted as factors contributing to carcinogenesis in the multiple subgroup (Table 4).

## Discussion

In this study, we hypothesized that the ABT, which accurately identifies ALDH2 inactivity noninvasively as a disease marker for superficial ESCC and superficial HPSCC treated with ESD, would exhibit clinical significance and usefulness. To support this hypothesis, we

**Table 2. Comparing risk factors between SCC and HC groups: Univariate analysis and multivariate analysis.**

| Characteristics | Subjects | | Univariate analysis | Multivariate analysis | | |
|---|---|---|---|---|---|---|
| | SCC group ($n = 100$) | HC group ($n = 275$) | $p$ | OR | 95% CI | $p$ |
| Gender (male/female) | 95/5 | 167/108 | <0.001 | | | |
| Mean age (years) | 66.4 ± 8.5 | 49.1 ± 10.9 | <0.001 | 1.189 | 1.130–1.251 | <0.001 |
| Mean daily alcohol consumption (g) | 53.6 ± 29.4 | 15.1 ± 18.9 | <0.001 | 1.079 | 1.056–0.102 | <0.001 |
| Mean HRA score | 8.17 ± 2.46 | 3.41 ± 2.87 | <0.001 | 1.585 | 1.322–1.900 | <0.001 |
| Toothbrushing score | 1.90 ± 1.35 | 1.32 ± 1.35 | <0.001 | | | |
| A/E ratio | 24.0 ± 16.5 | 18.8 ± 16.2 | <0.001 | 1.048 | 1.021–1.075 | <0.001 |

A/E, acetaldehyde-to-ethanol; CI, confidence interval; HC, healthy control; HRA, health risk appraisal; OR, odds ratio; SCC, squamous cell carcinoma.

**Table 3. Characteristics of patients with single SCC and multiple SCC.**

| Characteristics | Single (*n* = 63) | Multiple (*n* = 31) | *p* |
|---|---|---|---|
| Gender (male/female) | 60/3 | 29/2 | 0.731 |
| Age (years) | | | |
| 40–49 | 0% | 3.2% | |
| 50–59 | 19.0% | 12.9% | |
| 60–69 | 42.9% | 48.4% | |
| 70–79 | 34.9% | 25.8% | |
| ≥80 | 3.2% | 9.7% | 0.305 |
| Mean ± SD | 66.3 ± 8.2 | 66.6 ± 8.4 | 0.965 |
| Daily alcohol consumption (g) | | | |
| <25 | 11.1% | 12.9% | |
| ≥25 | 88.9% | 87.1% | 0.799 |
| Mean ± SD | 52.7 ± 26.3 | 55.9 ± 35.8 | 0.984 |
| Smoking | 28/35 | 13/18 | 0.818 |
| Alcohol flushing | | | |
| Current flushing | 27.0% | 29.0% | |
| Former flushing | 22.2% | 25.8% | |
| Never flushing | 50.8% | 45.2% | 0.869 |
| Alcohol flush reaction (yes/no) | 32/31 | 14/17 | 0.608 |
| Mean HRA score | 8.35 ± 2.19 | 7.81 ± 2.96 | 0.318 |
| Toothbrushing score | | | |
| 0–2 | 60.3% | 65.5% | |
| 3–4 | 34.5% | 32.2% | |
| 5–6 | 5.2% | 6.9% | 0.791 |
| Mean ± SD | 1.87 ± 1.85 | 1.84 ± 2.00 | 0.793 |
| Lugol-voiding lesions | | | |
| Grade A | 20.6% | 16.1% | |
| Grade B | 42.9% | 19.4% | |
| Grade C | 36.5% | 64.5% | 0.029 |
| Grades A+B/C | 40/23 | 11/20 | 0.010 |
| Cancer of other organs | | | |
| Yes/no | 13/50 | 5/26 | 0.602 |

The "single" subgroup comprised patients in whom a single lesion was initially treated with ESD; the "multiple" subgroup comprised patients in whom multiple lesions were initially treated with endoscopic submucosal dissection (ESD).

HRA, health risk appraisal.

demonstrated that ABT was useful for identifying patients with superficial ESCC or superficial HSCC and that the A/E ratio was useful for identifying patients with multiple lesions.

In this study, the proportions of men, heavy drinkers, and smokers appeared to be higher among the patients with SCC than among those with HC. In addition to alcohol consumption, smoking was reported to be a generally accepted major etiological factor of upper digestive tract cancers [31] As expected, the results were similar in this study.

Alcohol metabolic capacity has been found to vary greatly with regard to race, and 35% of the Japanese population have inactive ALDH2 [32]. Individuals with inactive ALDH2 who drink heavily are at high risk for ESCC and HPSCC [17–21]. Genetic testing is the most reliable method for identifying ALDH2*2 allele carriers and has become more widely used, and large-scale genotyping is a possibility for some cases. However, this may not be the case for all

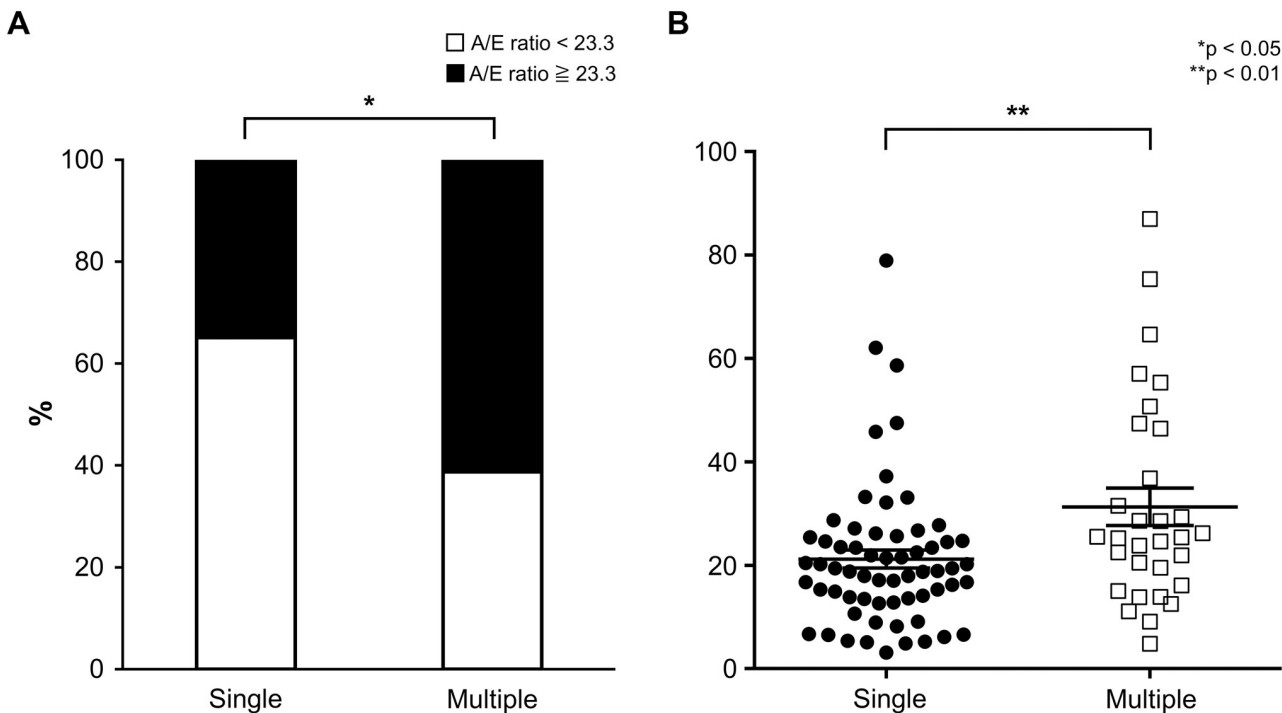

**Fig 3. Comparison of the A/E ratios between single subgroup and multiple subgroup.** (**a**) The proportion of individuals with inactive acetaldehyde dehydrogenase 2 (ALDH2; acetaldehyde-to-ethanol [A/E] ratio $\geq$ 23.3) was significantly higher among patients with Multiple subgroup (61.3%) than among those with Single subgroup (34.9%; $p$ = 0.015, chi-squared test). (**b**) The A/E ratio in patients with Multiple subgroup (31.3 ± 20.0) was significantly higher than that in patients with Single subgroup (21.2 ± 13.9; $p$ = 0.008, Mann–Whitney $U$ test).

the world due to the ethical aspects of genotyping. The alcohol flushing questionnaire and the ethanol patch test have been considered as alternative diagnostic tools. However, their accuracy is unsatisfactory [20,33]. The ABT could accurately identify ALDH2*2 allele carriers within 8 min; thus, people would be able to know the result shortly after the test [23].

The A/E ratio detected carriers of the ALDH2*2 allele with high sensitivity and specificity [23] Aoyama reported that the breath levels of acetaldehyde varied considerably even between participants with the same genotype, and showed an unsatisfactory diagnostic performance for identifying carriers of the ALDH2*2 allele. The results of this study showed a strong correlation between acetaldehyde and ethanol levels in each breath sample. The following reasons have been proposed for the mechanism of the increase in the A/E ratio in breath after alcohol administration. After alcohol consumption, ethanol was reported to be rapidly diffused to

**Table 4. Comparing risk factors between single and multiple subgroups: Univariate analysis and multivariate analysis.**

| Characteristics | Subgroup | | Univariate analysis | Multivariate analysis | | |
|---|---|---|---|---|---|---|
| | Single (*n* = 63) | Multiple (*n* = 31) | *p* | OR | 95% CI | *p* |
| Gender (male/female) | 60/3 | 29/2 | 0.731 | | | |
| Mean age (years) | 66.3 ± 8.2 | 66.6 ± 8.4 | 0.965 | | | |
| Lugol-voiding lesions | 40/23 | 11/20 | 0.010 | 2.682 | 1.060–6.788 | <0.001 |
| A/E ratio | 21.2 ± 13.9 | 31.3 ± 20.0 | 0.008 | 1.032 | 1.003–1.061 | <0.001 |

The "single" subgroup comprised patients in whom a single lesion was initially treated with ESD; the "multiple" subgroup comprised patients in whom multiple lesions were initially treated with endoscopic submucosal dissection (ESD). A/E, acetaldehyde-to-ethanol.

saliva covering the mucosal surfaces of the oral cavity, hypopharynx, and presumably, the oesophagus. This is followed by instant oxidation of ethanol locally to acetaldehyde [34] Conversely, after alcohol consumption, the salivary acetaldehyde concentration was approximately twofold higher in the ALDH2-deficients than in the ALDH2-actives [35]. The elevated A/E ratio in ALDH2-deficients may thus reflect the changes in acetaldehyde and ethanol concentrations in saliva and might have partly contributed to ALDH2 deficiency by acetaldehyde derived from the pulmonary blood to breath.

In recent years, early esophageal cancer was reported to have a good prognosis; several studies have shown that the overall rate of 5-year survival in patients with intramucosal ESCC who underwent endoscopic resection was 89%–95% [24,36,37]. Furthermore, the cause-specific rate of 5-year survival among patients with superficial pharyngeal cancer who underwent endoscopic resection was 97% [25]. ESD as a minimally invasive treatment for superficial pharyngeal cancer was reported to be effective and safe [38]. As mentioned previously, early detection of ESCC and HPSCC is very important for long-term survival of patients with esophageal cancer.

According to a recent report, multiple LVLs in the esophagus increase the risk of multiple SCC [38]. However, Lugol chromoendoscopy is invasive and painful or uncomfortable. A major advantage of ABT is its noninvasiveness. We demonstrated that the A/E ratio and LVL grade were extracted as factors contributing to carcinogenesis in the multiple subgroup; the ABT can be used easily in healthy patients at high risk of ESCC and HPSCC. Furthermore, conducting the ABT at a young age in healthy individuals may help prevent ESCC and HPSCC by encouraging these individuals to modify drinking habits.

On the other hand, in patients with ESCC, the incidence of multiple lesions simultaneously is high. In addition, 10% to 50% of patients with HPSCC also have ESCC [39–42]. It is extremely important to accurately diagnose HPSCC and ESCC that have occurred at the same time. Our results indicate that ABT, which reflects alcohol metabolic ability, may be useful for identifying such patients.

This study had some limitations. First, because genetic testing is time consuming and cumbersome, it was not conducted in this study. However, Aoyama et al. [23] reported that the A/E ratio could identify ALDH2*2 allele carriers very accurately (in their study, the rate of accuracy was 96.4%). Second, in this study, the weight of the patient was not considered. According to the Japanese law, the alcohol concentration in a soft drink must be within the range ($<$1%). In addition, the limit of the amount that the tested person could drink quickly was considered to be 100 mL. In the future, the amount of reagent might need to be considered according to body size. Third, bacteria within the mouth also produce a low acetaldehyde level [43]. In this study, acetaldehyde produced from oral bacteria could not be measured. However, the effect of aldehyde produced by oral bacteria is considered to be minute because the end expiratory air is collected in ABT. Fourth, patients who had undergone gastrointestinal surgery, which could have affected the breath ethanol or acetaldehyde levels, were not tested in this study. Fifth, the subjects of this study were patients who received only endoscopic treatment; we did not include patients who underwent surgery or chemotherapy for esophageal cancer and pharyngeal cancer. In the future, we also need to test the ABT in patients who underwent surgery and chemotherapy.

## Conclusion

ABT may be a useful screening tool for detecting people at risk of ESCC and HPSCC. In addition, ABT could be a useful tool for detecting patients at risk of multiple or double carcinomas

among those with ESCC and HPSCC. In the future, the use of ABT may help prevent pharyngeal and esophageal cancer by encouraging individuals to modify their drinking habits.

## Supporting information

**S1 File. Daily alcohol consumption, calculated as the sum of scores A to D for each age.**
(TIF)

**S2 File. Grading and appearance of Lugol-voiding lesions (LVLs).** The number of LVLs per endoscopic view was counted, and the grading was divided into three categories.
(TIF)

**S3 File. ABT data HC and SCC Kagoshima corrected version2 20210429.**
(XLS)

## Acknowledgments

The authors would like to thank Ms. Yuko Morinaga-Nakamura for her technical assistance.

## Author Contributions

**Conceptualization:** Fumisato Sasaki, Kohei Oda, Akio Ido.

**Data curation:** Fumisato Sasaki, Kohei Oda, Hidehito Maeda, Masayuki Kabayama, Hiromichi Iwaya, Yuga Komaki, Shiho Arima, Shiroh Tanoue, Hiroshi Fujita.

**Formal analysis:** Fumisato Sasaki.

**Visualization:** Kohei Oda, Yuga Komaki, Shinichi Hashimoto.

**Writing – original draft:** Fumisato Sasaki.

**Writing – review & editing:** Shuji Kanmura, Shiroh Tanoue, Akio Ido.

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
