## [Decision Letter · Decision Letter 0]

16 Mar 2021

PONE-D-21-01891

Aldehyde breath test as a disease marker in patients with esophageal and hypopharyngeal squamous cell carcinoma

PLOS ONE

Dear Dr. Sasaki,

Thank you for submitting your manuscript to PLOS ONE. After careful consideration, we feel that it has merit but does not fully meet PLOS ONE’s publication criteria as it currently stands. Therefore, we invite you to submit a revised version of the manuscript that addresses the points raised during the review process.

Please address all the comments and questions from the reviewers.

In addition, please clarify:

The authors stated that they used Cox proportional hazards regression. Shouldn’t it be logistic regression because of the case-control design the ORs instead or HRs were presented.The healthy control group was individuals that underwent medical checkup that included esophagogastroduodenoscopy. It was surprising that there was no difference in smoking between SCC and HC. Please comment on the potential selection bias of the control subjects.

We look forward to receiving your revised manuscript.

Kind regards,

Jeffrey S Chang

Academic Editor

PLOS ONE

Journal Requirements:

3. Please provide a sample size and power calculation in the Methods, or discuss the reasons for not performing one before study initiation.

Reviewers' comments:

Reviewer's Responses to Questions

**Comments to the Author**

1. Is the manuscript technically sound, and do the data support the conclusions?

Reviewer #1: Yes

Reviewer #2: Partly

Reviewer #3: Partly

2. Has the statistical analysis been performed appropriately and rigorously? 

Reviewer #1: Yes

Reviewer #2: Yes

Reviewer #3: Yes

3. Have the authors made all data underlying the findings in their manuscript fully available?

Reviewer #1: Yes

Reviewer #2: Yes

Reviewer #3: Yes

4. Is the manuscript presented in an intelligible fashion and written in standard English?

Reviewer #1: Yes

Reviewer #2: Yes

Reviewer #3: Yes

5. Review Comments to the Author

Reviewer #1: Sasaki and colleagues determined weather assessing a ratio of acetaldehyde/ethanol in the breath after an alcohol challenge (100mL) could be useful in predicting ESCC and HPSCC. Overall the manuscript is interesting and a few minor comments and suggestions.

1) How the breath test is expected to be used as a tool is a bit unclear and would suggest writing how you would implement this into the medical decision making process. I could see this as more of a screening tool possibly for rural areas of countries that do not routinely have medical care. I do not think that this could be considered a diagnostic tool to detect cancer (as it is more a secondary screen for inactive ALDH2) rather than cancer itself.

2) The ethanol dose chosen is not a weight based dose. This should be mentioned in the limitations and also how this was overcome by calculating an acetaldehyde/ethanol ratio.

3) Bacteria within the mouth also produce a low level of acetaldehyde. Particularly in people who do not often brush their teeth it would be important to speculate or show whether this level of acetaldehyde from bacteria may or may not impact the measurements of acetaldehyde.

4) I would suggest on page 5 the figure 1 table 1 and accompanying text be moved to the first part of the results section.

5) Since Plos One is a general interest journal, it would be helpful to readers to add a paragraph about the why people with an ALDH2*2 variant are more at risk for head and neck cancer. This article Annu Rev Pharmacol Toxicol. 2015; 55: 107–127 doi: 10.1146/annurev-pharmtox-010814-124915 discusses this well (see Table 1) as it shows how the activity of ALDH2 within the esophageal tissue is much lower relative to other gastrointestinal tissues. Adding inactivating ALDH2 genetics leads the environment of the head and neck more at risk for DNA damage, dysplasia and cancer.

6) page 14, line 240. The sentence starting with "Alcohol metabolic..." is unclear and would suggest rewording/rewriting the sentence.

7) page 14, line 243. The idea that genotyping is not suitable for mass screening is not entirely accurate. Perhaps for only certain populations. For example in the United States genotyping kids are now being sold on Amazon.com to detect ALDH2*2. Also the genetics company 23 and me provides whether you have the rs671 SNP as part of their basic report. So the idea of genotyping on a large scale is likely for some people but maybe not for all people. I would reword this sentence in addition to the other section this was mentioned within the manuscript.

Reviewer #2: In the paper by Fumisato Sasaki et al. acetaldehyde breath test (ABT) has been used as a disease marker in patients with esophageal and hypopharyngeal squamous cell carcinoma. The main aim of the study was to find a useful and non-invasive tool for the identification of individuals at high risk for pharyngeal and esophageal cancer, as well as for the detection of multiple carcinomas. It is suggested that the use of ABT may help to prevent pharyngeal and esophageal cancer by encouraging individuals to modify their drinking habits. The paper has some merit, but before its possible acceptance a major revision is recommended.

Major Comments

1. The results confirm in part the earlier findings of Aoyama et al. i.e., that ABT test may be a useful tool for the identification of ALDH2*2 carriers (ALDH2-deficient alcohol drinkers), who are at high risk for esophageal and head and neck cancers. However, the study is retrospective and thus does not give an answer to the most important question: Can ABT test used to encourage individuals to modify their drinking habits?

2. Alcohol flushing questionnaire and HRA score (Yokoyama et al. 2012) have been successfully used as rather simple screening tools for the detection of ALDH2-deficient and heavily drinking risk groups for esophageal cancer. In the present study the ABT test should be compared HRA score in order to find out does ABT test indeed provide any significant diagnostic improvement in the detection of esophageal and hypopharyngeal cancers.

3. The mechanism behind the elevated A/E-ratio in breath after alcohol administration has not been discussed at all. However, there is enough published data to make that possible. After alcohol drinking ethanol is rapidly diffused to saliva covering the mucosal surfaces of oral cavity, hypopharynx and presumably also oesophagus. This is followed by instant oxidation of ethanol locally to acetaldehyde. In ALDH2-actives this process is mediated almost exclusively by microbes representing normal oral flora. After alcohol drinking salivary acetaldehyde concentration is about two times higher in ALDH2-deficients than in ALDH2-actives. The additional acetaldehyde is delivered to the saliva most probably from the salivary glands.

In several studies the highly significant correlation between salivary ethanol and acetaldehyde levels has been confirmed. It should be noted that the evaporation point of acetaldehyde is 20.2 of Celsius. Thus, most if not all of breath acetaldehyde is derived from saliva by evaporation. This may be the case also with most of breath ethanol. The elevated acetaldehyde/ethanol ratio in ALDH2-deficients may thus reflect the changes in acetaldehyde and ethanol concentrations in saliva and in addition may in part be contributed in ALDH2-deficients by acetaldehyde derived from the pulmonary blood to the breath.

Minor notes

1. Acetaldehyde instead of aldehyde should be used both in the title and the main text.

2. If possible, ALDH2 genotyping would markedly improve the significance of the findings.

3. Tooth brushing scores: the range in methods is 0 – 2, but in the table 1 it is 0-6??

4. Patients with SCC appear to be more often men, heavy drinkers and smokers than patients with HC. Do the authors have any explanation?

Reviewer #3: Comments to the author, Manuscript #PONE-D-21-01891

Sasaki et al. described the application of an acetaldehyde breath test (ABT) to distinguish between ESD treated SCC patients and a health control group based on a cut-off value from the previously published acetaldehyde-to-ethanol (A/E) ratio. Higher AE value and the number of predicted ALDH2*2 subjects were higher in the SSC patient group. Furthermore, both parameters were also higher in the multiple SSC lesion group than in the single lesion group. The major claim of this study was the potential of using ABT as a disease marker for SSC and for differentiation between single and multiple lesions of ESCC and HPSCC. However, the reviewer finds these claims rather weak based on the data provided in this manuscript.

Major concerns and comments:

1. The lack of real ALDH2 genotype data for the subjects in this study. The ABT for ALDH2 genotype diagnosis as published by Aoyama et al. [22] has not been independently confirmed by any other research group. Such a confirmation ought to be carried out in this study. The authors claimed that “because genetic testing is time consuming and cumbersome, it was not conducted in this study”. This is not the case, with various currently available genotyping methods and techniques. Genotyping the number of samples described in this study (n<400) should not time consuming, cumbersome, nor expensive. The review strongly recommends that genotyping be carried out not only for ALDH2, but also for ADH1B. As it is well known that the ADH1B polymorphism (rs1229984) among the Japanese strongly influence the rate of conversation of alcohol to acetaldehyde, the alcohol flushing reaction and risks of ESCC and HPSCC. The information on genotype status from this study can confirm the validity of using A/E ratio and also taking into account role ADH1B gene.

2. There was very significant difference in gender and age distribution between the SSC group and health control group (e.g. male/female ration 95/5 vs 167/108 (Table 1)). It is well known that men and women metabolize alcohol differently. Hence the difference in A/E ratio, alcohol flushing reaction etc. could be simply due the gender difference in the subjects recruited. Same concern for the average age and age distribution described in the two groups. Another example: the lower toothbrushing frequency in the SSC group could simple be due to more men in this group than in the HC group. If men and are in general had worse oral hygiene habit than women, the gender difference can also explain the observed result and there was no real relationship for being more susceptible to SCC cancer. Recommendation: should find a gender, age matched HC group for this study.

3. Table 1 flushing reaction: why no statistical difference between the two groups? Individuals with ALDH2*2 genotype were expected (as shown) to be statistically higher in the SCC group than the HC group. Shouldn’t alcohol flushers should also be higher in the SSC group?

4. Higher A/E ratio in the multiple lesion group could be due to more subject with ALDH2*2 genotype and rather than due to multiple cancer per se. ALDH2*2 is defined by A/E ratio >23.3. Since there are more subjects with A/E ratio in the multiple lesion group, therefore higher A/E values in the multiple group vs. single cancer group is higher is naturally expected. The two are correlated. The A/E ratio from ABT at best is a good predictor of ALDH2 genotype.

In addition, this study did not show unequivocally that it can be used to distinguished between single and multiple cancer in a clinical setting, since the overlap of the two groups was significant as show in the figures. In other words, from these data, it only be drawn to conclude that A/E ratio is a good predictor of ALDH2*2 genotypes. The claim of this paper aldehyde breath test can be use as a disease marker in patients with esophageal and hypopharyngeal squamous cell carcinoma is there weak. The authors need to demonstrate that ABT can distinguish cancer per se, not the genotype.

Minor mistakes and comments:

1. Page 20, Incomplete sentence: “Alcohol metabolic capacity has been found to vary greatly with regard to race, and 35% of the Japanese population [30].”

2. The “single subgroup” (n = 61) written in the abstract should be (n=63). The reviewer also subject change the total number of SSC patients to 94, rather than 100 as stated in abstract, since 6 patients were excluded for the study. This will keep the numbers more consistent.

3. What is HRA, health risk appraisal? Need to add a few more sentences to describe some details on what elements in HRA for the readers were.

6. PLOS authors have the option to publish the peer review history of their article (what does this mean?). If published, this will include your full peer review and any attached files.

Reviewer #1: No

Reviewer #2: No

Reviewer #3: No

---

## [Author Response · Author response to Decision Letter 0]

6 Apr 2021

PONE-D-21-01891

Aldehyde breath test as a disease marker in patients with esophageal and hypopharyngeal squamous cell carcinoma

PLOS ONE

=>I ensured that my manuscript meets PLOS ONE's style requirements, including those for file naming. 

=> I corrected above. All relevant data are within the manuscript and its Supporting Information files.

=> I corrected above. All relevant data are within the manuscript and its Supporting Information files. 

=> I corrected above. All relevant data are within the manuscript and its Supporting Information files. 

3. Please provide a sample size and power calculation in the Methods, or discuss the reasons for not performing one before study initiation.

=> Since this is a cross-sectional study, sample size and power were not calculated performing one before study initiation.

Reviewers' comments:

Reviewer's Responses to Questions

Comments to the Author

1. Is the manuscript technically sound, and do the data support the conclusions?

Reviewer #1: Yes

Reviewer #2: Partly

Reviewer #3: Partly

2. Has the statistical analysis been performed appropriately and rigorously?

Reviewer #1: Yes

Reviewer #2: Yes

Reviewer #3: Yes

3. Have the authors made all data underlying the findings in their manuscript fully available?

Reviewer #1: Yes

Reviewer #2: Yes

Reviewer #3: Yes

4. Is the manuscript presented in an intelligible fashion and written in standard English?

Reviewer #1: Yes

Reviewer #2: Yes

Reviewer #3: Yes

 

5. Review Comments to the Author

Reviewer #1: Sasaki and colleagues determined weather assessing a ratio of acetaldehyde/ethanol in the breath after an alcohol challenge (100mL) could be useful in predicting ESCC and HPSCC. Overall the manuscript is interesting and a few minor comments and suggestions.

1) How the breath test is expected to be used as a tool is a bit unclear and would suggest writing how you would implement this into the medical decision making process. I could see this as more of a screening tool possibly for rural areas of countries that do not routinely have medical care. I do not think that this could be considered a diagnostic tool to detect cancer (as it is more a secondary screen for inactive ALDH2) rather than cancer itself.

Thank you for your suggestion. We agree with your opinion. We consider ABT to be a tool to identify groups at high risk for pharyngeal and esophageal cancer, not a tool to point out cancer itself. We have corrected conclusions.

 “ABT may be screening tool possibly to detect people has risk for ESCC and HPSCC. In addition, ABT could be a useful tool to detect patients has risk for multiple or double carcinomas for ESCC and HPSCC patients”.

2) The ethanol dose chosen is not a weight based dose. This should be mentioned in the limitations and also how this was overcome by calculating an acetaldehyde/ethanol ratio.

Thank you for your suggestion, and the following sentence has been added to Limitation

“Second, in this study, the weight of the patient was not considered. According to Japanese law, the alcohol concentration as a soft drink must be within the range (less than 1%). In addition, the limit of the amount that the tested person could drink quickly was considered to be 100 mL. In the future, it may be necessary to consider the amount of reagent according to body size.”

3) Bacteria within the mouth also produce a low level of acetaldehyde. Particularly in people who do not often brush their teeth it would be important to speculate or show whether this level of acetaldehyde from bacteria may or may not impact the measurements of acetaldehyde.

We agree with your opinion. We have added the following text. “In this study, acetaldehyde produced from oral bacteria could not be measured. However, the effect of aldehyde produced from oral bacteria is considered to be minute because the end expiratory air is collected in ABT

4) I would suggest on page 5 the figure 1 table 1 and accompanying text be moved to the first part of the results section.

We agree with your opinion. We moved to the first part of the results section the figure 1 table 1 and accompanying text.

5) Since Plos One is a general interest journal, it would be helpful to readers to add a paragraph about the why people with an ALDH2*2 variant are more at risk for head and neck cancer. This article Annu Rev Pharmacol Toxicol. 2015; 55: 107–127 doi: 10.1146/annurev-pharmtox-010814-124915 discusses this well (see Table 1) as it shows how the activity of ALDH2 within the esophageal tissue is much lower relative to other gastrointestinal tissues. Adding inactivating ALDH2 genetics leads the environment of the head and neck more at risk for DNA damage, dysplasia and cancer.

We agree with your opinion. We have added the following text.

“It has been reported that the reason why people with ALDH2*2 variants have a higher risk of head and neck cancer and esophageal cancer is because the activity of ALDH2 in their tissues is much lower relative to other gastrointestinal tissues.[＊This article Annu Rev Pharmacol Toxicol. 2015; 55: 107–127 doi: 10.1146/annurev-pharmtox-010814-124915]”

6) page 14, line 240. The sentence starting with "Alcohol metabolic..." is unclear and would suggest rewording/rewriting the sentence.

We agree with your opinion. I have corrected it.

“Alcohol metabolic capacity has been found to vary greatly with regard to race, and 35% of the Japanese population being ALDH2 inactive. [30].”

7) page 14, line 243. The idea that genotyping is not suitable for mass screening is not entirely accurate. Perhaps for only certain populations. For example in the United States genotyping kids are now being sold on Amazon.com to detect ALDH2*2. Also the genetics company 23 and me provides whether you have the rs671 SNP as part of their basic report. So the idea of genotyping on a large scale is likely for some people but maybe not for all people. I would reword this sentence in addition to the other section this was mentioned within the manuscript.

We agree with your opinion. I have corrected it.

“Genetic testing is the most reliable way to identify ALDH2*2 allele carriers and becoming more widespread, and large-scale genotyping is a possibility for some.”

 

Reviewer #2: In the paper by Fumisato Sasaki et al. acetaldehyde breath test (ABT) has been used as a disease marker in patients with esophageal and hypopharyngeal squamous cell carcinoma. The main aim of the study was to find a useful and non-invasive tool for the identification of individuals at high risk for pharyngeal and esophageal cancer, as well as for the detection of multiple carcinomas. It is suggested that the use of ABT may help to prevent pharyngeal and esophageal cancer by encouraging individuals to modify their drinking habits. The paper has some merit, but before its possible acceptance a major revision is recommended.

Major Comments

1. The results confirm in part the earlier findings of Aoyama et al. i.e., that ABT test may be a useful tool for the identification of ALDH2*2 carriers (ALDH2-deficient alcohol drinkers), who are at high risk for esophageal and head and neck cancers. However, the study is retrospective and thus does not give an answer to the most important question: Can ABT test used to encourage individuals to modify their drinking habits?

Thank you for your suggestion. This study did not examine whether the ABT test can be used to encourage individuals to improve their drinking habits. We are planning to conduct a prospective, multicenter study in patients with esophageal and pharyngeal cancers. We are planning to conduct a prospective, multicenter study in esophageal and pharyngeal cancer patients, and we would like to clarify the results in the study.

2. Alcohol flushing questionnaire and HRA score (Yokoyama et al. 2012) have been successfully used as rather simple screening tools for the detection of ALDH2-deficient and heavily drinking risk groups for esophageal cancer. In the present study the ABT test should be compared HRA score in order to find out does ABT test indeed provide any significant diagnostic improvement in the detection of esophageal and hypopharyngeal cancers.

Thank you for your suggestion. In the present study, both ABT and HRA scores were independent risk factors for esophageal and hypopharyngeal cancer in multivariate analysis (Table 2). Therefore, we consider both of them as risk factors for esophageal and hypopharyngeal cancer.

3. ¬¬ The mechanism behind the elevated A/E-ratio in breath after alcohol administration has not been discussed at all. However, there is enough published data to make that possible. After alcohol drinking ethanol is rapidly diffused to saliva covering the mucosal surfaces of oral cavity, hypopharynx and presumably also oesophagus. This is followed by instant oxidation of ethanol locally to acetaldehyde. In ALDH2-actives this process is mediated almost exclusively by microbes representing normal oral flora. After alcohol drinking salivary acetaldehyde concentration is about two times higher in ALDH2-deficients than in ALDH2-actives. The additional acetaldehyde is delivered to the saliva most probably from the salivary glands.

In several studies the highly significant correlation between salivary ethanol and acetaldehyde levels has been confirmed. It should be noted that the evaporation point of acetaldehyde is 20.2 of Celsius. Thus, most if not all of breath acetaldehyde is derived from saliva by evaporation. This may be the case also with most of breath ethanol. The elevated acetaldehyde/ethanol ratio in ALDH2-deficients may thus reflect the changes in acetaldehyde and ethanol concentrations in saliva and in addition may in part be contributed in ALDH2-deficients by acetaldehyde derived from the pulmonary blood to the breath.

Thank you for your suggestion. We have added the following text. 

“The A/E ratio was able to detect carriers of the ALDH2*2 allele with high sensitivity and specificity [Clinical and Translational Gastroenterology (2017) 8, e96;]. Aoyama reported that breath acetaldehyde levels varied considerably even between participants with the same genotype, and their diagnostic performance for identifying carriers of the ALDH2*2 allele was unsatisfactory. The results of the study showed that they observed a strong correlation between the acetaldehyde and ethanol levels in each breath sample. The following reasons have been proposed for the mechanism of the increase in the A/E ratio in in breath after alcohol administration.　After alcohol drinking ethanol was reported that rapidly diffused to saliva covering the mucosal surfaces of oral cavity, hypopharynx and presumably also oesophagus. This is followed by instant oxidation of ethanol locally to acetaldehyde. [Lekarski. 2011 Jan;30(175):69-74.] In other hands, after alcohol drinking salivary acetaldehyde concentration is about two times higher in ALDH2-deficients than in ALDH2-actives.[ Clin Exp Res. 2000 Jun;24(6):873-7.]The elevated acetaldehyde/ethanol ratio in ALDH2-deficients may thus reflect the changes in acetaldehyde and ethanol concentrations in saliva and in addition may in part be contributed in ALDH2-deficients by acetaldehyde derived from the pulmonary blood to the breath.”

Minor notes

1. Acetaldehyde instead of aldehyde should be used both in the title and the main text.

Thank you for your suggestion. We have made the change.

2. If possible, ALDH2 genotyping would markedly improve the significance of the findings.

Thank you for your suggestion. In this study, we did not perform ALDH2 genotyping because the A/E ratio could identify ALDH2*2 allele carriers very accurately and this study was a cross-sectional study. We are planning to conduct a multicenter, prospective study including ALDH2 genotyping in this study.

3. Tooth brushing scores: the range in methods is 0 – 2, but in the table 1 it is 0-6??

Thank you for your suggestion. The tooth brushing score is 0 - 2 for morning, noon and night respectively, for a total of 0 - 6 points per day.

4. Patients with SCC appear to be more often men, heavy drinkers and smokers than patients with HC. Do the authors have any explanation?

Thank you for your suggestion. I have added the following text to the Discussion.

“In this study, patients with SCC appear to be more often men, heavy drinkers and smokers than patients with HC. In addition to alcohol consumption, smoking was reported that also a generally accepted major etiological factor for upper digestive tract cancers[Pelucchi C, Gallus S, Garavello W, Bosetti C, La Vecchia C (2008) Alcohol and tobacco use, and cancer risk for upper aerodigestive tract and liver. Eur J Cancer Prev 17: 340-344. doi:10.1111/j. 1365-2354.2007.00865.x. PubMed: 18562959.]　As expected, the results were similar in this study.”

 

Reviewer #3: Comments to the author, Manuscript #PONE-D-21-01891

Sasaki et al. described the application of an acetaldehyde breath test (ABT) to distinguish between ESD treated SCC patients and a health control group based on a cut-off value from the previously published acetaldehyde-to-ethanol (A/E) ratio. Higher AE value and the number of predicted ALDH2*2 subjects were higher in the SSC patient group. Furthermore, both parameters were also higher in the multiple SSC lesion group than in the single lesion group. The major claim of this study was the potential of using ABT as a disease marker for SSC and for differentiation between single and multiple lesions of ESCC and HPSCC. However, the reviewer finds these claims rather weak based on the data provided in this manuscript.

Major concerns and comments:

1. The lack of real ALDH2 genotype data for the subjects in this study. The ABT for ALDH2 genotype diagnosis as published by Aoyama et al. [22] has not been independently confirmed by any other research group. Such a confirmation ought to be carried out in this study. The authors claimed that “because genetic testing is time consuming and cumbersome, it was not conducted in this study”. This is not the case, with various currently available genotyping methods and techniques. Genotyping the number of samples described in this study (n<400) should not time consuming, cumbersome, nor expensive. The review strongly recommends that genotyping be carried out not only for ALDH2, but also for ADH1B. As it is well known that the ADH1B polymorphism (rs1229984) among the Japanese strongly influence the rate of conversation of alcohol to acetaldehyde, the alcohol flushing reaction and risks of ESCC and HPSCC. The information on genotype status from this study can confirm the validity of using A/E ratio and also taking into account role ADH1B gene.

Thank you for your suggestion. We agree with your opinion. In this study, we did not perform ALDH2 genotyping because the A/E ratio could identify ALDH2*2 allele carriers very accurately and this study was a cross-sectional study. We are planning to conduct a multicenter, prospective study including ADH1B and ALDH2 genotyping in next study. In other hands, we have found a new significance in measuring and evaluating the A/E ratio numerically, rather than simply using it for genotyping. In this study (Table.2), the A/E ratio as a continuous variable was extracted as a significant factor in the multivariate analysis, and we believe that the effect of acetaldehyde on carcinogenesis can be evaluated by quantifying it, taking into account individual differences, rather than simply dividing the subjects into two groups by genotyping. In the future, we will be able to evaluate the effect of acetaldehyde on carcinogenesis. Furthermore, in the future, we plan to use ABT to screening tests based on the risk of carcinogenesis and to use the results as a tool for lifestyle guidance.

Aoyama reported in previous study that the breath ethanol level was not　influenced by the ADH1B genotype, although breath ethanol　levels in carriers of the ADH1B*1/*1 genotype are theoretically　higher than those in carriers of the ADH1B*2 allele (ADH1B*1/*2　and ADH1B*2/*2). We assume that this is because of the limited　difference in ADH1B activity between ADH1B*1 and ADH1B*2　alleles. The breath acetaldehyde level also was not influenced　by the ADH1B genotype (data not shown). These results are　consistent with previous reports that ADH1B genotype does not　correlate with blood ethanol or acetaldehyde levels after　ingestion of usual amounts of alcohol.

2. There was very significant difference in gender and age distribution between the SSC group and health control group (e.g. male/female ration 95/5 vs 167/108 (Table 1)). It is well known that men and women metabolize alcohol differently. Hence the difference in A/E ratio, alcohol flushing reaction etc. could be simply due the gender difference in the subjects recruited. Same concern for the average age and age distribution described in the two groups. Another example: the lower toothbrushing frequency in the SSC group could simple be due to more men in this group than in the HC group. If men and are in general had worse oral hygiene habit than women, the gender difference can also explain the observed result and there was no real relationship for being more susceptible to SCC cancer. Recommendation: should find a gender, age matched HC group for this study.

Thank you for your suggestion. We agree with your opinion. We would have liked to have conducted this study in an age- and gender-matched population. However, we were not able to conduct this study in a population matched for age and gender, because many of the patients in Japan are younger than the general population, and the elderly often receive hospital care through insurance. We would like to clarify this in future prospective studies.

3. Table 1 flushing reaction: why no statistical difference between the two groups? Individuals with ALDH2*2 genotype were expected (as shown) to be statistically higher in the SCC group than the HC group. Shouldn’t alcohol flushers should also be higher in the SSC group?

Thank you for your suggestion. We agree with your opinion. Never Flushing was 48% in the HC group and 59% in the SCC group, and although the difference was not statistically significant, there was a trend toward a higher percentage of Never Flushing in the SCC group. The difference may become significant if the number of cases increases.

4. Higher A/E ratio in the multiple lesion group could be due to more subject with ALDH2*2 genotype and rather than due to multiple cancer per se. ALDH2*2 is defined by A/E ratio >23.3. Since there are more subjects with A/E ratio in the multiple lesion group, therefore higher A/E values in the multiple group vs. single cancer group is higher is naturally expected. The two are correlated. The A/E ratio from ABT at best is a good predictor of ALDH2 genotype.

In addition, this study did not show unequivocally that it can be used to distinguished between single and multiple cancer in a clinical setting, since the overlap of the two groups was significant as show in the figures. In other words, from these data, it only be drawn to conclude that A/E ratio is a good predictor of ALDH2*2 genotypes. The claim of this paper aldehyde breath test can be use as a disease marker in patients with esophageal and hypopharyngeal squamous cell carcinoma is there weak. The authors need to demonstrate that ABT can distinguish cancer per se, not the genotype.

Thank you for your suggestion. We agree with your opinion. We consider ABT to be a tool to identify groups at high risk for pharyngeal and esophageal cancer, not a tool to point out cancer itself. We have corrected conclusions.

 “ABT may be screening tool possibly to detect people has risk for ESCC and HPSCC. In addition, ABT could be a useful tool to detect patients has risk for multiple or double carcinomas for ESCC and HPSCC patients”. In other hands, we have found a new significance in measuring and evaluating the A/E ratio numerically, rather than simply using it for genotyping. In this study (Table.2), the A/E ratio as a continuous variable was extracted as a significant factor in the multivariate analysis, and we believe that the effect of acetaldehyde on carcinogenesis can be evaluated by quantifying it, taking　into account individual differences, rather than simply dividing the subjects into two groups by genotyping. In the future, we will be able to evaluate the effect of acetaldehyde on carcinogenesis. Furthermore, in the future, we plan to use ABT to screening tests based on the risk of carcinogenesis and to use the results as a tool for lifestyle guidance.

Minor mistakes and comments:

1. Page 20, Incomplete sentence: “Alcohol metabolic capacity has been found to vary greatly with regard to race, and 35% of the Japanese population [30].”

We agree with your opinion. I have corrected it.

Alcohol metabolic capacity has been found to vary greatly with regard to race, and 35% of the Japanese population being ALDH2 inactive. [30].

2. The “single subgroup” (n = 61) written in the abstract should be (n=63). The reviewer also subject change the total number of SSC patients to 94, rather than 100 as stated in abstract, since 6 patients were excluded for the study. This will keep the numbers more consistent.

Thank you for your suggestion. We agree with your opinion. I have corrected it. Six of the 100 patients in the SCC group were not first-time ESD patients, so they were excluded from the subanalysis of patients (single group vs. multiple group), but were included in the study of HC group vs. SCC group.

3. What is HRA, health risk appraisal? Need to add a few more sentences to describe some details on what elements in HRA for the readers were.

Thank you for your suggestion. We agree with your opinion. I have corrected it. 

“The HRA score is useful for assessing the risk of esophageal cancer in Japanese. It is calculated from the amount and frequency of alcohol consumption, smoking, flushing, and consumption of fruits and vegetables. The higher the score, the higher the risk.”

---

## [Decision Letter · Decision Letter 1]

19 Apr 2021

PONE-D-21-01891R1

Acetaldehyde breath test as a disease marker in patients with esophageal and hypopharyngeal squamous cell carcinoma

PLOS ONE

Dear Dr. Sasaki,

Thank you for submitting your manuscript to PLOS ONE. After careful consideration, we feel that it has merit but does not fully meet PLOS ONE’s publication criteria as it currently stands. Therefore, we invite you to submit a revised version of the manuscript that addresses the points raised during the review process.

Please address the minor comments from the reviewers.

We look forward to receiving your revised manuscript.

Kind regards,

Jeffrey S Chang

Academic Editor

PLOS ONE

Journal Requirements:

Reviewers' comments:

Reviewer's Responses to Questions

**Comments to the Author**

1. If the authors have adequately addressed your comments raised in a previous round of review and you feel that this manuscript is now acceptable for publication, you may indicate that here to bypass the “Comments to the Author” section, enter your conflict of interest statement in the “Confidential to Editor” section, and submit your "Accept" recommendation.

Reviewer #1: All comments have been addressed

Reviewer #2: (No Response)

2. Is the manuscript technically sound, and do the data support the conclusions?

Reviewer #1: Yes

Reviewer #2: Yes

3. Has the statistical analysis been performed appropriately and rigorously? 

Reviewer #1: Yes

Reviewer #2: Yes

4. Have the authors made all data underlying the findings in their manuscript fully available?

Reviewer #1: Yes

Reviewer #2: Yes

5. Is the manuscript presented in an intelligible fashion and written in standard English?

Reviewer #1: Yes

Reviewer #2: No

6. Review Comments to the Author

Reviewer #1: All comments were addressed. One suggestion would be to revise this introductory sentence for the relative activity of the ALDH2*1/*2 genotype- from the literature the activity is somewhere between 40-60% relative to wild type and <10% activity for a heterozygote relative to wild type ALDH2 seems to be a bit high.

"ALDH2 genotypes are classified as follows: ALDH2*1/*1 (homozygous active ALDH2); ALDH2*1/*2 (heterozygous inactive [<10% activity] ALDH2; and ALDH2*2/*2 (homozygous inactive [0% activity] ALDH2) [10-13]"

Reviewer #2: Minor comments:

Row 1/Title: Acetaldehyde breath test as a disease marker... Should be: ...as a cancer risk marker....

Row 53: The latter prognosis word should be deleted

Row 59: ...acetaldehyde dehydrogenase 2 (ALDH2) should be aldehyde dehydrogenase 2

Row 271: Reference number 34 is a review article and written in polish. Better to use the original article: Linderborg K et al. Food Chem Toxicol 2011,49:2103-06.

7. PLOS authors have the option to publish the peer review history of their article (what does this mean?). If published, this will include your full peer review and any attached files.

Reviewer #1: No

Reviewer #2: No

---

## [Author Response · Author response to Decision Letter 1]

22 Apr 2021

Jeffrey S Chang

Academic Editor

PLOS ONE

Manuscript ID: PONE-D-21-01891

Manuscript title: Acetaldehyde breath test as a cancer risk marker in patients with esophageal and hypopharyngeal squamous cell carcinoma

Authors: Fumisato Sasaki et al.

Dear Editor:

Thank you for your letter dated April 19 2021. We are pleased to know that our manuscript has been rated as potentially acceptable for publication in PLOS ONE, subject to adequate revision and response to the comments raised by the reviewers.

Based on the instructions provided in the decision letter and comments provided by the reviewers, we have revised the manuscript by modifying the relevant sections of the manuscript. We have uploaded a copy of the original manuscript marked with all the changes made during the revision process. The revisions made are shown highlighted in yellow. Also appended to this letter are point-by-point responses to the comments raised by the reviewers. 

We would like to take this opportunity to express our sincere thanks to the reviewers who identified areas of the manuscript that needed corrections or modification. We would like also to thank you for allowing us to resubmit a revised copy of the manuscript.

We hope that the revised manuscript is accepted for publication in PLOS ONE.

Sincerely Yours,

Fumisato Sasaki, M.D., Ph.D

Review Comments to the Author

Reviewer #1: All comments were addressed. One suggestion would be to revise this introductory sentence for the relative activity of the ALDH2*1/*2 genotype- from the literature the activity is somewhere between 40-60% relative to wild type and <10% activity for a heterozygote relative to wild type ALDH2 seems to be a bit high.

=> Thank you for your suggestion.I searched for papers on the activity of ALDH2*1/*2 and found two new papers reporting that ALDH2*1/*2 (heterozygous inactive form) is less than 20%. References 12 and 13 have been replaced.I was not able to find any papers that reported that ALDH2*1/*2 (heterozygous inactive form) is 40-60%.

Reviewer #2: Minor comments:

Row 1/Title: Acetaldehyde breath test as a disease marker... Should be: ...as a cancer risk marker....

=>Thank you for pointing this out. The title has been changed.

Row 53: The latter prognosis word should be deleted

=> Thank you for your suggestion. I deleted the latter prognosis word.

Row 59: ...acetaldehyde dehydrogenase 2 (ALDH2) should be aldehyde dehydrogenase 2

Row 271: Reference number 34 is a review article and written in polish. Better to use the original article: Linderborg K et al. Food Chem Toxicol 2011,49:2103-06.

=> Thank you for your suggestion. I corrected the points you raised.

---

## [Editor Report · Decision Letter 2]

27 Apr 2021

Acetaldehyde breath test as a cancer risk marker in patients with esophageal and hypopharyngeal squamous cell carcinoma

PONE-D-21-01891R2

Dear Dr. Sasaki,

We’re pleased to inform you that your manuscript has been judged scientifically suitable for publication and will be formally accepted for publication once it meets all outstanding technical requirements.

Kind regards,

Jeffrey S Chang

Academic Editor

PLOS ONE
---

## [Editor Report · Acceptance letter]

7 May 2021

PONE-D-21-01891R2 

Acetaldehyde breath test as a cancer risk marker in patients with esophageal and hypopharyngeal squamous cell carcinoma 

Dear Dr. Sasaki:

I'm pleased to inform you that your manuscript has been deemed suitable for publication in PLOS ONE. Congratulations! Your manuscript is now with our production department. 

Kind regards, 

on behalf of

Dr. Jeffrey S Chang 

Academic Editor

PLOS ONE